# An Established HPLC-MS/MS Method for Evaluation of the Influence of Salt Processing on Pharmacokinetics of Six Compounds in Cuscutae Semen

**DOI:** 10.3390/molecules24132502

**Published:** 2019-07-09

**Authors:** Jiao Liu, Shuhan Zou, Wei Liu, Jin Li, Hui Wang, Jiao Hao, Jun He, Xiumei Gao, Erwei Liu, Yanxu Chang

**Affiliations:** 1Tianjin State Key Laboratory of Modern Chinese Medicine, Tianjin University of Traditional Chinese Medicine, Tianjin 300193, China; 2Tianjin Key Laboratory of Phytochemistry and Pharmaceutical Analysis, Tianjin University of Traditional Chinese Medicine, Tianjin 300193, China

**Keywords:** Cuscutae Semen, salt processed, HPLC-MS/MS, pharmacokinetic study

## Abstract

A sensitive and effective method was developed for clarifying the pharmacokinetic properties of six compounds (including hyperin, chlorogenic acid, neochlorogenic acid, p-coumaric acid, astragalin, and isoquercitrin) in two processed Cuscutae Semen samples by high performance liquid chromatography mass spectrometry (HPLC-MS/MS). The six compounds were separated by acetonitrile and 0.1% formic acid-water on an Agilent Eclipse plus C18 column (4.6 mm × 100 mm, 1.8 μm). All compounds were analyzed with negative ion mode in multiple reaction monitoring (MRM). The lower limits of quantification (LLOQ) of hyperin, astragalin, neochlorogenic acid, chlorogenic acid, isoquercitrin, and p-coumaric acid were 1, 0.1, 4, 0.1, 2, and 4 ng·mL^−1^, respectively. The validated approach was effectively used for the pharmacokinetics of six compounds of two processed Cuscutae Semen samples after oral administration to rat. The results indicated that salt processing could improve the adsorption and bioavailability of astragalin in Cuscutae Semen.

## 1. Introduction

Cuscutae Semen (CS) derived from the dry and mature seeds of *Cuscuta australis* R.Br. or *Cuscuta chinensis* Lam. is named TuSiZi in Chinese [1]. It is used to nourish the kidney, improve sexual function, treat impotence, prevent miscarriage, and restrain diarrhea [2,3]. CS contains some chemical components, including flavonoids, chlorogenic acids, alkaloids, and polysaccharides [4,5]. CS possesses many biological activities, such as anti-inflammatory, neuroprotective, and antinociceptive effects [6,7]. In recent years, some research has indicated that flavonoids and chlorogenic acids are the main active ingredients in CS for anti-inflammatory and antioxidant effects, for preventing miscarriage, and for cardiovascular activities [8,9,10,11].

Processing of traditional Chinese medicine (TCM) is a pharmaceutical method to satisfy the needs of treatment. According to the theory of TCM, medicinal materials need special processing methods, such as stir-frying, steaming, calcining, or boiling, which aim to increase effects and decrease toxicity. According to Chinese pharmacopoeia [2], there are two different processing methods of CS, including the raw Cuscutae Semen (R-CS) processed by stir frying and stir frying with salt-water, named the stir-frying Cuscutae Semen (SF-CS), and salt-processed Cuscutae Semen (SP-CS), respectively. The SF-CS and SP-CS have different pharmacological properties and are named qing-chao-tu-si-zi and yan-zhi-tu-si-zi in Chinese, respectively. when the medicinal materials are processed, the chemical compounds change, including the increase or decrease of content and the formation or disappearance of compounds. According to the basic theories of TCM, medicinal materials processed with salt can improve the function of nourishing the kidney [12]. The previous report indicated that the contents of quercetin and total compounds increased and hyperoside decreased after salt processing. SP-CS also has enhanced levels of testosterone and antioxidant effects from a biological activity aspect [13]. However, the work mechanism of activity change was not clear after salt processing in vivo.

Pharmacokinetics is an effective method of evaluating the safety of medicine and to clarify the influence of processing on the components absorbed into the body after oral administration in rats [14]. At present, there are only a few reports for pharmacokinetic studies of two compounds (hyperoside and quercetin) after oral administration of Cuscutae Semen extract [15,16]. Firstly, the pharmacokinetic information of the two compounds did not clarify the metabolic regularity of the whole herbs in vivo. Next, there has been no report comparing the pharmacokinetic studies of two processing products for CS. Therefore, it is necessary to perform the pharmacokinetics of multiple flavonoid ingredients to clarify the work mechanism of activity change by comparing the pharmacokinetic properties of two processing procedures for CS.

In the present research, an efficient and validated method was developed for simultaneous quantification of six compounds, including hyperin, chlorogenic acid, neochlorogenic acid, p-coumaric acid, astragalin, and isoquercitrin (Figure 1) in pharmacokinetics of CS. Furthermore, the study of multiple compounds in rat plasma were measured in the pharmacokinetic study of two processing products for CS. The established method was effectively used to provide the pharmacokinetic outlines for clinic research of salt-processed CS.

## 2. Results

### 2.1. Internal Standard (IS) Selection

In the study, the three kinds of compounds and the three kinds of phenolic acids were simultaneously determined. Formononetin was selected as the internal standard for both flavonoid and phenolic acid owing to the similar structure and it showing no interference from endogenous substances.

### 2.2. Optimization of LC-MS/MS Conditions

In terms of MS conditions, the responses of analytes were better in multiple-reaction monitoring (MRM) with the negative ion mode. The optimized MS conditions are shown in Table 1. When acetonitrile and 0.1% formic acid-water were chosen with a flow rate of 0.5 mL·mL^−1^, the six compounds obtained a better separation and higher response.

### 2.3. Quantification of the Six Compounds in Two Processed CS Extracts

According to the extraction yield, the contents of the six compounds for the two processed CS were analyzed. The oral concentrations of hyperin, chlorogenic acid, neochlorogenic acid, p-coumaric acid, astragalin, and isoquercitrin were 21.8, 31.1, 6.09, 0.59, 1.87, and 1.98 mg·kg^−1^ for SF-CS extracts, respectively, and were 27.5, 49.0, 6.64, 0.87, 1.30, and 1.99 mg·kg^−1^ for SP-CS extracts, respectively.

### 2.4. Method Validations

#### 2.4.1. Linearity and LLOQ

The LLOQ, correlation coefficients and equations of the calibration curves of the six compounds are listed in Table 2. The linearity range was 1 to 250 ng·mL^−1^ for hyperin, from 0.1 to 1500 ng·mL^−1^ for chlorogenic acid, 4 to 1000 ng·mL^−1^ for neochlorogenic acid, 4 to 10,000 ng·mL^−1^ for p-coumaric acid, 0.1 to 25 ng·mL^−1^ for astragalin, and 2 to 500 ng·mL^−1^ for isoquercitrin. Each correlation coefficient was greater than 0.9990 (r ≥ 0.9990). The lower limits of quantification (LLOQ) of hyperin, neochlorogenic acid, p-coumaric acid, chlorogenic acid, astragalin, and isoquercitrin was 1, 4, 4, 0.1, 0.1, and 2 ng·mL^−1^, respectively. The accuracy was within ranged from 90.4 to 104% and relative standard deviation (RSD) (*n* = 6) was less than 17.4%.

#### 2.4.2. Selectivity

Selectivity was confirmed by determining the blank plasma from six different lots and comparing the MRM chromatographic profiles of plasma samples, which spiked with the six compounds. As shown in Figure 2, there was a good separation and no interference from endogenous components for all analytes.

#### 2.4.3. Accuracy and Precision

As shown in Table 3, the precision was below 17.0% and the accuracy range of quality control (QC) samples was 80.3 to 114% at three levels for intra-day and inter-day values. It was demonstrated that the present method was reproducible and precise.

#### 2.4.4. Stability

As shown in Table 4, the range of accuracy was 88.8 to 119%, while the RSD was within 15.1% for the stock solution of analytes. This indicated that the six compounds were stable under conditions of three freeze-thaw, being kept in auto-sampler for 24 h, and stored at −80 °C for 1 month.

#### 2.4.5. Matrix Effects and Recoveries

As listed in Table 3, the recoveries of the six compounds ranged from 60.0% to 117%, while the RSD was below 12.0% at three concentration levels. Moreover, the matrix effects of the six compounds were in the range of 89.0% to 117%, while the RSD was less than 14.0% at three concentrations.

### 2.5. Pharmacokinetic Application

After oral administration of the two processed products of CS, the only one that could not be tested (isoquercitrin) for the six compounds in rat plasma and the LLOQ reached 2 ng·mL^−1^. The established method was used to investigate the pharmacokinetics of the five compounds after oral administration of SF-CS and SP-CS extracts. The pharmacokinetic profiles of five compound were represented in a one-compartment model. The mean plasma concentration time-curve outliers are illustrated in Figure 3. As shown in Table 5, the AUC_(0–24h)_ of p-coumaric (3198 ± 635 and 2567 ± 792 ng·mL^−1^) was high in all compounds of the two processing procedures of CS, which indicated that it possessed abundant plasma exposure. The AUC_(0–24h)_ and C_max_ of hyperin and astragalin were lower than those of other compounds in the two processing procedures of CS, which demonstrated that absorption of them was low in vivo. Moreover, the neochlorogenic acid and chlorogenic acid exhibited the double-peak phenomenon, which is related to the transformation among the phenolic acids.

Comparing the pharmacokinetic parameters of SF-CS and SP-CS (Figure 4 and Table 5), the T_max_ of hyperin (0.74 ± 0.63 h) after oral administration of SF-CS extract was longer than that of hyperin (0.11 ± 0.04 h) after oral administration of SP-CS extract. The same trends for the other four components were found after oral administration of SF-CS and SP-CS extract. These results indicated that salt processing accelerated the absorption of the five compounds. Furthermore, the C_max_, AUC_0–24_, and AUC_0–∞_ of neochlorogenic acid of SF-CS have remarkable differences (*p* < 0.05 for SP-CS, which shows that the absorption of neochlorogenic acid can be improved in SF-CS. Moreover, AUC_(0–24h)_ of astragalin was 1.27 ± 0.76 and 1.31 ± 0.27 ng·mL^−1^ for the two processing procedures of CS, respectively. The C_max_ of astragalin for SF-CS had significance differences (*p* < 0.05 with SP-CS). It was demonstrated that the plasma exposure of astragalin in CS was increased after salt processing. The present study indicated that salt processing could significantly increase the bioavailability of astragalin and accelerate the adsorption of hyperin, neochlorogenic acid, chlorogenic acid, and p-coumaric acid. This could be because salt processing increased the solubility of astragalin.

In order to explore the effect of salt in SP-CS on bioavailability and adsorption of five compounds in SF-CS, the pharmacokinetics of the five compounds after oral administration of SF-CS extract and SF-CS extract with salt were also performed by LC-MS/MS method. Comparing the pharmacokinetic parameters of SF-CS and SF-CS with salt (Figure 5), there were no differences between the main parameters of SF-CS and SF-CS with salt. It was indicated that the salt in SP-CS has no effects on the bioavailability and absorption of the five compounds for CS. However, salt processing significantly increased the bioavailability of astragalin and accelerated the adsorption of hyperin, neochlorogenic acid, chlorogenic acid, and p-coumaric acid. The reason might be that salt processing changed the composition proportions of herbal extract and pharmacokinetic interactions of multiple components in vivo. The presumption needs to be verified by the further research.

## 3. Materials and Methods

### 3.1. Chemicals and Reagents

Reference substances of hyperin, neochlorogenic acid, chlorogenic acid, p-coumaric acid, astragalin, isoquercitrin, and formononetin (IS) with purity over 98.0% were achieved from Chengdu Must Biotechnology Co., Ltd. (Chengdu, China). The acetonitrile and formic acid were obtained from Dikma (Foothill Ranch, CA, USA). Ultra-pure water was provided by Milli-Q system (Millipore, Boston, MA, USA).

### 3.2. Apparatus and LC-MS/MS Conditions

All components were measured on an Agilent 1200 HPLC system (Agilent Corporation, Santa Clara, CA, USA) and API 3200 triple quadrupole mass spectrometer (Concord, ON, Canada) in negative ion mode. Acetonitrile (A) and 0.1% formic acid-water (B) were used to separate the analytes on an Agilent Eclipse plus C18 column (4.6 mm × 100 mm, 1.8 μm) with a flow rate of 0.5 mL·min^−1^. The gradient program was as follows: 17% A (0–10 min), 17–48% A (10–12.5 min), 48–52% A (12.5–14 min), 52–54% A (14–17 min), 54–60% A (17–18.5 min), 60–90% A (18.5–20 min). The equilibration time (10 min) of the sequence was added before the next injection. The column temperature was 30 °C. The optimized main parameters, namely curtain gas, collision gas, ion Spray voltage, and temperature, were set at 30 psi, 8 psi, −4000 V, and 600 °C, respectively. All of parameters are shown in Table 1.

### 3.3. Preparation of SF-CS and SP-CS Extract

The stir-frying and salt processing protocols for CS were performed according to the 0213 general rules recorded in the Chinese pharmacopeia 2015 edition [2].

The raw CS was heated with constant stirring until it became yellow with a slight crackle and light fragrance in a pot. Then, the SF-CS was poured out onto a plate to cool for the following experiment.

The raw CS was mixed well with salt water (2 g of common salt was dissolved in 100 mL of water) until they were infused thoroughly in a closed vessel. Next, they were heated with constant stirring at 100–110 °C for 4 min until they became yellow with a slight crackle and light fragrance [17]. Finally, the SP-CS was poured out onto a plate to cool for the following experiment.

SF-CS (1.5 kg) and SP-CS (1.5 kg) were accurately weighed, followed by addition of extraction solvent and refluxing twice independently (95% ethanol the first time and 60% ethanol the second time) with material-solvent ratio of 1:6 for 1 h each time. Under the reduced pressure, the extracted solutions were concentrated by using a rotary evaporator at 40 °C until dry. Extraction rates of SF-CS and SP-CS were 14.9% and 17.6%, respectively.

### 3.4. Preparation of Standard and Quality Control (QC) Samples

Stock solutions (1 mg·mL^−1^) were achieved by dissolving each standard substance in methanol. They were applied to obtain working solutions for calibration curves and QC samples by serial dilution. The IS solution was prepared in methanol at a concentration of 10 ng·mL^−1^. All solutions were stored at 4 °C.

### 3.5. Plasma Samples Preparation

The plasma sample (100 μL), IS (10 μL) and formic acid (10 μL) were mixed and vortexed for 1 min. Then, 1 mL of ethyl acetate was spiked into the mixed solution and centrifuged for 10 min at 14,000 rpm. The supernatant was transferred and evaporated with nitrogen gas. These residues were re-dissolved with 100 μL methanol. The final solutions were vortexed for 2 min and centrifuged for 10 min. The supernatant was injected into the HPLC-MS/MS and analyzed.

### 3.6. Method Validation

The established method was evaluated following USFDA guideline [18]. The parameters contained sensitivity, accuracy, precision, recovery and so on.

#### 3.6.1. Linearity and LLOQ

The calibration curve was performed by adding standard solutions into blank plasma. The concentration ranges of the standard solution were 0.1–1500, 0.1–25, 1–250, 2–500, 4–1000, and 4–10,000 ng·mL^−1^ for chlorogenic acid, astragalin, hyperin, isoquercitrin, neochlorogenic acid, and p-coumaric acid, respectively. The linearity was assessed with the peak area rate of the six compounds to IS concentration using 1/X^2^ weighting.

#### 3.6.2. Selectivity

The selectivity was performed by analyzing the blank plasma sample, blank plasma added into the six compounds at the concentration of LLOQ, and real plasma orally administered with the SF-CS and SP-CS extracts.

#### 3.6.3. Accuracy and Precision

The QC samples from six different batches at three levels (low, medium, and high) were applied to evaluate the accuracy and precision intra-day and over three different days. The assessed index was RSD and the percent ratios of the calculated concentration to nominal concentration. The range of accuracy should be 85.0 to 115%.

#### 3.6.4. Stability

The stabilities (including freeze-thaw cycles, auto-sampler and long-term stability) of all analytes were assessed with QC samples at three concentrations. All solutions were kept at 4 °C.

#### 3.6.5. Matrix Effects and Recoveries

The recovery and matrix effects of the six compounds were measured with six batches at three concentrations. The recovery was achieved by dividing the peak areas of the six compounds and IS added into plasma before extraction with those of the six compounds and IS added into processed plasma after extraction. The matrix effects were obtained by dividing results of the six compounds and IS added into processed plasma after extraction by those of the six compounds and IS in solvent. The values of recovery and matrix effects ranged from 85.0 to 115%.

### 3.7. Pharmacokinetic and Data Analysis

The pharmacokinetic study was conducted in accordance with the Guidelines for the Care and Use of Laboratory Animals by USA National Institutes of Health and approved by the Animal Ethics Committee of Tianjin University of Traditional Chinese Medicine (the permit number: TCM-LAEC2019023, Tianjin, China). Male Sprague-Dawley rats (200–240 g) were fed in the animal laboratory of Tianjin University of Traditional Chinese Medicine under the standard conditions. These rats were fed and given water for 1 week before experimentation. Before administration of these extracts, all rats were fasted, although they were still given water until 12 h prior. The extraction yield of medicinal materials was influenced by the process. In order to investigate the influence of processing on the metabolic process of active ingredients in vivo, the same amount of raw CS was processed and administrated. Depending on the clinical dosage that was applied to the human body, the dosage of SF-CS extracts was 1.9 g kg^−1^ (an equivalent dosage of 12.75 g kg^−1^ raw CS) for oral administration, while extraction of SP-CS was 2.2 g kg^−1^ (an equivalent dosage of 12.50 g kg^−1^ raw CS). In order to investigate the influence of salt on the metabolic process of active ingredients in vivo, the dosage of SF-CS extracts was 1.9 g kg^−1^ (an equivalent dosage of 12.75 g kg^−1^ raw CS) and SF-CS extracts with salt ( 0.26 g kg^−1^ ) were used to perform the pharmacokinetic study. Plasma (about 250 µL) was gathered in 1.5 mL heparinized polythene tubes at 0, 0.033, 0.083, 0.25, 0.5, 0.75, 1, 1.5, 2, 4, 6, 8, 12, and 24 h after oral administration of the two extracts. The collected plasma was centrifuged for 10 min at 6000 rpm and stored at −20 °C before analysis.

All of the pharmacokinetic parameters were estimated with the DAS1.0 system (Medical College of Wannan, China). The maximum concentration of oral administration of CS (C_max_) and the time to reach C_max_ (T_max_) were achieved from the concentration–time profile. All values were shown as mean ± standard deviation. Independent t-test was used to compare the pharmacokinetic parameters of the two processed CS.

## 4. Conclusions

A sensitive and efficient method was developed for simultaneous quantification of hyperin, chlorogenic acid, neochlorogenic acid, p-coumaric acid, and astragalin after oral administration of SF-CS and SP-CS extracts to rats. Moreover, the pharmacokinetic information revealed that salt processing accelerated the adsorption of hyperin, neochlorogenic acid, chlorogenic acid, and p-coumaric acid, and improved the bioavailability of astragalin. In conclusion, this investigation was established to assess pharmacokinetics of multiple orally administrated compounds of SF-CS and SP-CS extracts to rats. From a pharmacokinetic perspective, this can help us to understand the absorption of active compounds in vivo for two processed CS samples and will help in further study of SP-CS in clinical trials.

## Figures and Tables

**Figure 1 molecules-24-02502-f001:**
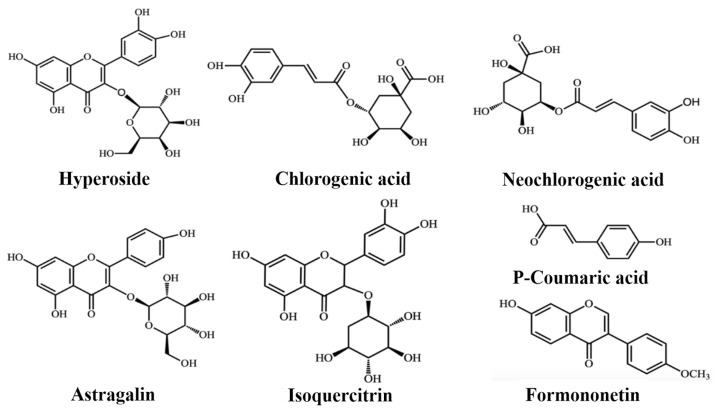
Chemical structures of hyperoside, chlorogenic acid, neochlorogenic acid, astragalin, isoquercitrin, P-Coumaric acid, and formononetin (IS).

**Figure 2 molecules-24-02502-f002:**
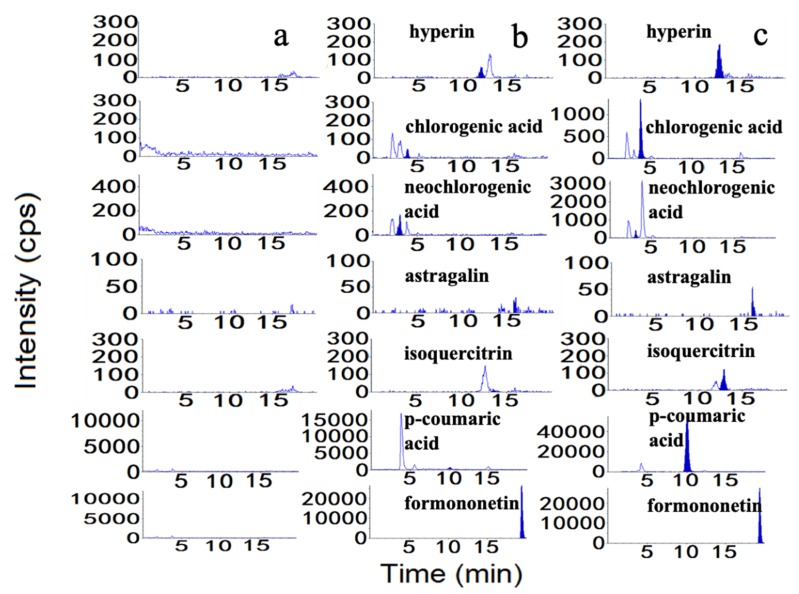
Representative chromatogram of (**a**) blank plasma, (**b**) blank plasma spiked with standard compounds at LLOQs, and (**c**) plasma sample.

**Figure 3 molecules-24-02502-f003:**
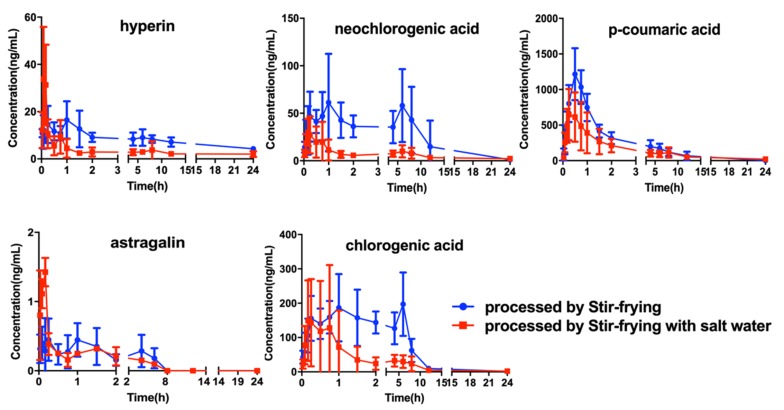
The mean plasma concentration time profiles of hyperin, neochlorogenic acid, chlorogenic acid, p-coumaric acid, and astragalin after oral administration of SF-CS and SP-CS (*n* = 10, mean ± SD).

**Figure 4 molecules-24-02502-f004:**
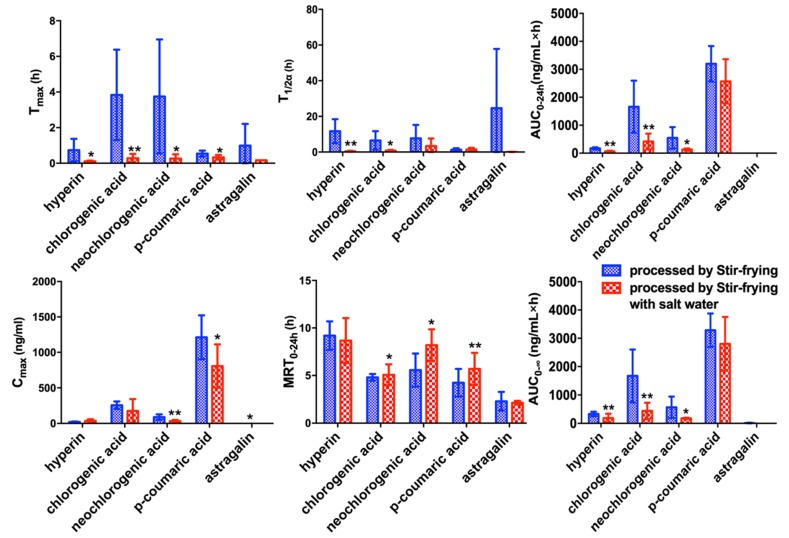
Pharmacokinetic comparison of oral administration of SF-CS and SP-CS extract (* *p* < 0.05, ** *p* < 0.01).

**Figure 5 molecules-24-02502-f005:**
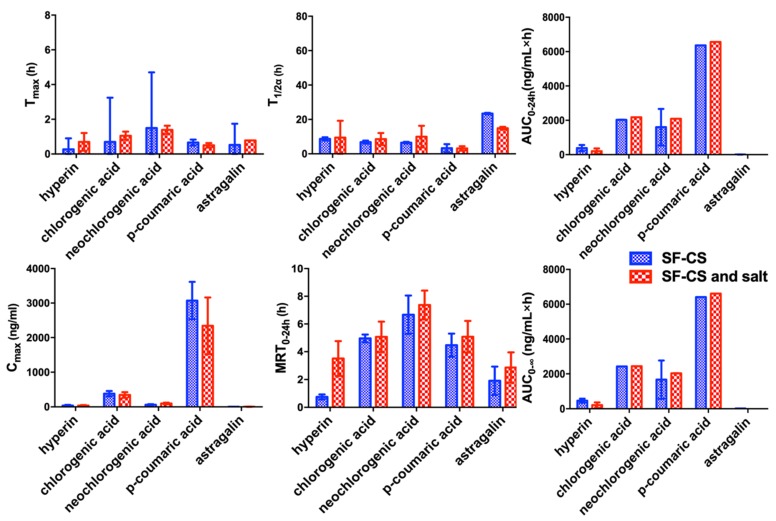
Pharmacokinetic comparison of oral administration of SF-CS extract and SF-CS extract and salt.

**Table 1 molecules-24-02502-t001:** The source parameters of the six compounds and internal standard (IS).

Compounds	Q_1_	Q_3_	Dwell Time (ms)	DP (V)	EP (V)	CE (eV)	CXP (V)	Retention (min)
Neochlorogenic acid	353.0	190.9	100	−40	−4.5	−30	−3.0	3.01
Chlorogenic acid	353.0	191.0	100	−40	−5.5	−25	−1.0	3.83
P-coumaric acid	162.8	119.0	100	−35	−5.5	−20	−2.0	10.07
Hyperin	463.0	300.0	100	−70	−6.0	−39	−1.0	12.51
Isoquercitrin	462.9	300.0	100	−60	−7.0	−40	−1.5	13.61
Astragalin	447.1	284.0	100	−68	−5.5	−38	−1.0	15.81
Formononetin (IS)	267.0	251.8	100	−60	−2	−27	−16	19.25

**Table 2 molecules-24-02502-t002:** Linear range, correlation coefficient(*r*), accuracy, and LLOQ (*n* = 6) employed in determination of the six compounds in rat plasma.

Compounds	Regression Equation	*r*	Linearity Range (ng·mL^−1^)	LLOQ (ng·mL^−1^)	Accuracy (%)	RSD (%)
Hyperin	Y = 0.00406X + 0.000488	0.9996	1–250	1	104	7.9
Chlorogenic acid	Y = 0.00113X + 0.00233	0.9992	0.1–1500	0.1	104	15
Neochlorogenic acid	Y = 0.00134X + 0.00254	0.9990	4–1000	4	102	4.6
P-coumaric acid	Y = 0.00366X + 0.00276	0.9993	4–10000	4	98.3	4.7
Astragalin	Y = 0.00613X + 0.000874	0.9991	0.1–25	0.1	90.4	17
Isoquercitrin	Y = 0.00286X + 0.000841	0.9994	2–500	2	102	10

**Table 3 molecules-24-02502-t003:** Intra-day and inter-day values, recovery, matrix effect accuracy, and precision of six compounds (*n* = 6).

Compounds	Concentration (ng·mL^−1^)	Intra-day		Inter-day		Recovery		Matrix Effect	
		Accuracy (%)	RSD (%)	Accuracy (%)	RSD (%)	Accuracy (%)	RSD (%)	Accuracy (%)	RSD (%)
Hyperin	3.0	94.8	12.7	98.9	4.20	93.0	6.0	95.0	5.0
	10	106	7.80	108	3.70	83.0	5.0	91.0	5.0
	250	107	12.5	100	6.40	87.0	8.0	92.0	7.0
Chlorogenic acid	0.3	114	14.7	106	7.10	65.0	11.0	109	13.0
	60	88.9	6.00	94.2	6.30	75.0	9.0	106	10.0
	1500	101	5.50	90.2	11.0	77.0	8.0	112	7.0
Neochlorogenic acid	12	103	3.70	99.3	3.90	76.0	8.0	117	7.0
	40	80.3	5.80	98.3	16.2	66.0	7.0	95.0	6.0
	1000	104	7.80	101	3.40	60.0	8.0	99.0	8.0
P-coumaric acid	12	108	1.80	97.2	10.0	61.0	6.0	101	9.0
	400	81.0	5.00	101	17.0	96.0	5.0	95.0	4.0
	10000	113	5.90	95.2	15.9	96.0	6.0	96.0	4.0
Astragalin	0.3	87.0	12.9	96.2	9.30	117	9.0	117	14.0
	1.0	113	9.50	112	8.00	94.0	11.0	99.0	8.0
	25	105	12.8	96.1	8.60	91.0	7.0	89.0	5.0
Isoquercitrin	6.0	86.0	12.7	93.5	6.00	95.0	9.0	100	8.0
	20	109	7.80	105	5.70	90.0	12.0	93.0	4.0
	500	110	13.2	100	10.9	89.0	8.0	92.0	7.0

**Table 4 molecules-24-02502-t004:** Stability of six compounds (*n* = 6).

Compounds	Concentration (ng·mL^−1^)	Freeze-Thaw Cycles		−80 °C for 1 Month		Auto-Sampler for 24 h	
		Accuracy (%)	RSD (%)	Accuracy (%)	RSD (%)	Accuracy (%)	RSD (%)
Hyperin	3.0	104	14.8	103	6.20	90.2	13.6
	10	116	10.7	92.5	3.80	107	13.2
	250	107	12.0	102	3.90	100	10.8
Chlorogenic acid	0.3	107	13.1	104	6.70	94.4	12.0
	60	119	12.6	105	13.5	115	13.9
	1500	104	14.2	99.8	10.1	119	9.20
Neochlorogenic acid	12	99.9	12.4	103	12.3	98.9	13.3
	40	102	8.30	103	14.3	115	9.00
	1000	116	8.10	90.6	9.50	117	4.90
P-coumaric acid	12	98.8	10.4	91.5	12.7	88.8	15.1
	400	112	8.30	100	11.9	116	8.30
	10000	111	4.20	93.6	8.40	115	8.60
Astragalin	0.3	91.7	12.1	113	12.8	96.0	9.90
	1.0	104	11.9	103	12.6	92.0	12.7
	25	102	12.8	92.9	12.0	95.2	12.7
Isoquercitrin	6.0	107	11.7	106	12.1	111	11.5
	20	109	14.2	91.4	13.2	103	8.00
	500	114	14.9	99.1	8.90	104	8.00

**Table 5 molecules-24-02502-t005:** Pharmacokinetic parameters of six compounds after oral administration of SF-CS and SP-CS extract (*n* = 10, mean ± SD).

Compound	Hyperin		Chlorogenic Acid		Neochlorogenic Acid		P-coumaric Acid		Astragalin		Isoquercitrin	
	A	B	A	B	A	B	A	B	A	B	A	B
dosage (mg/kg)	21.8	27.5	31.1	49.0	6.09	6.64	0.59	0.87	1.87	1.30	1.98	1.99
T_max_ (h)	0.74	0.11	3.84	0.28	3.75	0.26	0.54	0.33	0.99	0.17		
C_max_ (ng/mL)	21.3 ± 6.98	36.6 ± 21.9	257 ± 53	175.2 ± 167.7	88.6 ± 39.6	31.8 ± 17.3 **	1213 ± 310	805 ± 307 *	0.65 ± 0.39	1.43 ± 0.21 *	-	-
T_1/2ka_ (h)	2.15 ± 4.21	0.01 ± 0.01	0.63 ± 0.87	0.02 ± 0.01	1.12 ± 1.66	0.02 ± 0.02	0.15 ± 0.12	0.05 ± 0.04	2.27 ± 5.13	0.04 ± 0.00	-	-
T_1/2α_ (h)	11.7 ± 6.73	0.46 ± 0.30 **	6.53 ± 5.20	0.84 ± 0.42 *	7.72 ± 7.50	3.35 ± 4.31	1.32 ± 0.86	1.40 ± 0.95	24.6 ± 33.2	0.10 ± 0.04	-	-
AUC_0–24h_ (ng/mL·h)	172 ± 38	63.0 ± 27.8 **	1662 ± 931	417 ± 281 **	548 ± 384	129 ± 38 *	3198 ± 635	2567 ± 792	1.27 ± 0.76	1.31 ± 0.27	-	-
AUC_0–∞_ (ng/mL·h)	331 ± 80	185 ± 153 *	1671 ± 931	438 ± 289 **	567 ± 376	178 ± 22 *	3286 ± 591	2804 ± 951	11.2 ± 10.8	2.33 ± 0.72	-	-
MRT_0–24h_ (h)	9.19 ± 1.50	8.66 ± 2.37	4.81 ± 0.35	5.06 ± 1.10 *	5.57 ± 1.74	8.18 ± 1.66 *	4.24 ± 1.45	5.68 ± 1.69 **	2.30 ± 0.98	2.13 ± 0.20	-	-

Note: A = SF-CS; B = SP-CS; * *p* < 0.05, ** *p* < 0.01.

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
