# Peer review of "An Established HPLC-MS/MS Method for Evaluation of the Influence of Salt Processing on Pharmacokinetics of Six Compounds in Cuscutae Semen"

_molecules, 2019, doi:10.3390/molecules24132502_

Round 1

Reviewer 1 Report

How was the method validated when the signal to noise criteria for LLOQwas not met. The quantification of astragalin using the current method is unsatisfactory. 

To one of the reviewer's previous comment, the authors write that the noise is from metabolites in figure 2b. That is difficult to comprehend as one won't expect the metabolism in plasma after spiking of compounds (MS is specific). 

Fig 3. Error (SD) around the mean should in both the directions. Please, correct.  

Line 192-193: Give the reference for the pharmacopeia. What was the stirring speed? Slight crackle and light fragrance are scientifically unquantifiable. Please provide tangible extract preparation parameters which are reproducible. 

Line 197: Give a description of common salt. 

Reviewer 2 Report

Please address the experiment recommended by me in the previous review:

"please illustrate experimentally that the salt of the SF-CS isolated individual components have comparable PK properties to SFSW-CS."

Reviewer 3 Report

This manuscript looks unique and interesting to audience who are working on TCM with various processing methods and I would recommend to get it published without further revision.

Author Response

thank you for your review and approval.

Round 2

Reviewer 2 Report

The manuscript meets the minimum criteria for publication.

This manuscript is a resubmission of an earlier submission. The following is a list of the peer review reports and author responses from that submission.

Round 1

Reviewer 1 Report

It is unclear what authors imply when they write "two processed products of CS". Please elaborate and change accordingly in the manuscript to avoid confusion to the readers. 

How was the does selected for dosing the animals for PK study? Please briefly explain.

Briefly discuss the reasons behind the potential reason for the drop in accuracy as compared to the other compounds?

Why is noise higher than the analyte peak for most compounds in Fig 2? Isn't the interference at LLOQ an indication of endogenous noise with the extraction method? 

The peak for astragalin is almost missing at LLOQ and tested concentration. Also, correct the spelling in C.

Please give a similar number of significant figures for different values in the tables. 

Please explain what authors mean by line 133-134. 

Please specify what does the number and error for each PK parameter represent. Is it mean, SD, average, SEM?

Also, Tmax should be expressed as median as it's not a continuous variable. 

Fig 4. An error exists on both sides. Please, correct. Also, include information about the values which bar graphs represent in the legends. Similarly, change the table 4 legends. 

Materials and methods are poorly described so as to make it almost impractical for other researchers to reproduce the study results. Complete sample preparation process needs to be provided (use supplemental information if the word limit is exceeded in the manuscript).

What salt was used? What temperature was used for drying the extracts?

Give the concentration of IS in the extraction mixture.

How were the losses int he sample preparation accounted for? How was it ascertained that the extraction process is efficient and authors are not losing the analytes during the process?

Give the ethics clearance committee approval number for the study.

Plasma is not plural.

How many rats per samples and per time collection were used?

Reviewer 2 Report

Dear Authors,

The manuscript is of modest interest and requires additional experiments to support the conclusions. As proof-of-concept: please illustrate experimentally that the salt of the SF-CS isolated individual components have comparable PK properties to SFSW-CS.

Additionally, improved PK profiles does not necessarily mean that SFSW-CS have better  or even comparable clinical efficiency to SF-CS as stated in the conclusion. In vitro or In vivo studies are needed to confirm the same.

Also, extensive editing of English (grammar, spelling, and scientific terminology) is very much required.